# Hearing Rehabilitation with Cochlear Implants after CyberKnife Radiosurgery of Vestibular Schwannoma: A Report Based on Four Clinical Cases

**DOI:** 10.3390/brainsci11121646

**Published:** 2021-12-14

**Authors:** Sophia M. Häußler, Agnieszka J. Szczepek, Stefan Gräbel, Carolin Senger, Franziska Löbel, Markus Kufeld, Heidi Olze

**Affiliations:** 1Department of Otorhinolaryngology, Head and Neck Surgery, Campus Virchow-Klinikum, Charité-Universitätsmedizin Berlin, Corporate Member of Freie Universität Berlin and Humboldt Universität zu Berlin, 13353 Berlin, Germany; stefan.graebel@charite.de (S.G.); heidi.olze@charite.de (H.O.); 2Department of Otorhinolaryngology, Head and Neck Surgery, Campus Charité Mitte, Charité-Universitätsmedizin Berlin, Corporate Member of Freie Universität Berlin and Humboldt Universität zu Berlin, 10117 Berlin, Germany; agnes.szczepek@charite.de; 3Charité CyberKnife Center, Department of Radiation Oncology, Charité Universitätsmedizin Berlin, Corporate Member of Freie Universität Berlin, Humboldt-Universität zu Berlin, Berlin, Germany, Charité Universitätsmedizin, 13353 Berlin, Germany; carolin.senger@charite.de (C.S.); franziska.loebel@charite.de (F.L.); 4European CyberKnife Center München-Grosshadern, 81377 Munich, Germany; markus.kufeld@cyber-knife.net

**Keywords:** cochlear implant, CyberKnife, radiosurgery, vestibular schwannoma

## Abstract

Severe sensorineural hearing loss can be a symptom of the benign tumor vestibular schwannoma (VS). The treatment of VS with non-invasive stereotactic radiosurgery (SRS) offers a high local tumor control rate and an innovative possibility of sequential hearing rehabilitation with cochlear implantation. This study evaluated the feasibility, complications, and auditory outcomes of such a therapeutic approach. Three males and one female (mean age 65.3 ± 9.4 years) scheduled for cochlear implantation and diagnosed with sporadic VS classified as T1 or T2 (according to Samii) were enrolled in this study. All patients had progressive hearing loss qualifying them for cochlear implantation. First, the tumor was treated using CyberKnife SRS. Next, sequential auditory rehabilitation with a cochlear implant (CI) was performed. Clinical outcomes and surgical feasibility were analyzed, and audiological results were evaluated using pure tone audiometry and speech recognition tests. All patients exhibited open-set speech understanding. The mean word recognition score (at 65 dB SPL, Freiburg Monosyllabic Test, FMT) improved after cochlear implantation in all four patients from 5.0 ± 10% (with hearing aid) preoperatively to 60.0 ± 22.7% six months postoperatively. Our results suggest that in patients with profound hearing loss caused by sporadic vestibular schwannoma, the tumor removal with SRS followed by cochlear implantation is an effective method of auditory rehabilitation.

## 1. Introduction

In rare cases, vestibular schwannoma (VS) can induce profound hearing loss or deafness, a condition qualifying the patients for cochlear implantation. VS are benign neoplasms originating from Schwann cells surrounding the 8th cranial nerve as a myelin sheath [1]. The overall incidence of VS in the general population is about 1.7–4.2 per 100,000 [2,3]. Schwannomas can develop intracanalicularly (IC), in the cistern of the cerebellopontine angle (CPA), or intralabyrinthine. Intralabyrinthine schwannoma (ILS) and VS are increasingly detected with cranial magnetic resonance imaging (cMRI) [4,5]. VS accounts for approximately 3.0–3.4% [6,7] of the causes of sudden SHL. 

For patients with deafness or profound hearing impairment, cochlear implantation offers a possibility for auditory rehabilitation. A treatment plan should be established before implantation if an incidental finding of VS’s or ILS’s occurs during the cMRI evaluation. According to the accepted guidelines and recently published studies [8,9,10,11], the therapeutic options for the treatment of schwannoma include microsurgical resection, fractionated radiotherapy, or stereotactic radiosurgery (SRS) with either Gamma Knife, linear accelerator (LINAC) or CyberKnife; another option is the watch-and-scan observation management. 

SRS for the treatment of VS was introduced in the 1980s by Lecksell [12]. Since then, it has gained importance as a minimally invasive, single session technique with a low risk of side effects compared to microsurgery [13]. The CyberKnife is an image-guided robotic SRS system with a 6-MV linear accelerator on a six-axis robotic arm. That flexible arm allows multiple, non-isocentric irradiation angles to gain optimal coverage of even irregularly shaped tumors. Therefore, the tumor can be irradiated precisely without needing safety margins, reducing the surrounding tissue exposure. Additionally, there is a high rate of long-term tumor control, as shown by Windisch et al. [14]. They demonstrated 90.8% 10-year estimated local tumor control in their study of more than 1000 patients with VS.

Hearing loss is a common symptom of VS’s, but the reasons for hearing loss in VS patients are still not fully understood. The widely accepted view assumes dysfunction of or damage to the eighth cranial nerve or its peripheral branches. This hypothesis is supported by the pathological findings in auditory brainstem response and the cochlear nerve atrophy [15,16,17,18]. Nevertheless, pathological otoacoustic emissions and histopathological abnormalities such as a degeneration of the stria vascularis and the organ of Corti suggest that VS might also account for the cochlear changes [19,20,21]. Cochlear implantation for hearing rehabilitation in VS patients was first performed in patients with neurofibromatosis type 2 (NF2) and bilateral deafness, or those with VS in the only hearing ear [22,23,24,25]. There are also a few reports concerning cochlear implantation of patients who became deaf after SRS for VS [26,27,28], but mostly in NF2 patients with bilateral deafness [29]. 

This report discusses sporadic VS, concomitant functional deafness, CyberKnife treatment, and sequential CI hearing rehabilitation. The innovative concept of SRS and CI for patients with sporadic VS and hearing loss is presented and evaluated regarding feasibility and auditory outcomes.

## 2. Materials and Methods

The local Ethics Committee approved this retrospective study (approval number: EA2/030/13), which was conducted at a tertiary referral center to review VS treatment and hearing rehabilitation methods. All patients signed written informed consent. 

Three males and one female (mean age 65.3 ± 9.4 years) were retrospectively included in this study. The patients were diagnosed with profound sensorineural hearing loss and referred to our Cochlear Implantation Unit by secondary care specialists. All patients expressed their intention to undergo sequential hearing rehabilitation after SRS. “Sequential” was defined as cochlear implantation performed weeks, months, or years after SRS. All patients were diagnosed with sporadic VS. Neurofibromatosis was excluded on clinical and molecular pathology levels. 

Clinical diagnostics included collecting focused medical history, an ear, nose and throat (ENT) examination, vestibular examination, and bedside head impulse test (bHIT). In addition, videonystagmography and caloric testing were performed. Audiological performance was tested with pure tone audiometry (PTA), and the word recognition score WRS performed with Freiburg Monosyllabic Test (FMT) with numbers and monosyllables in quiet at 65 dB SPL (sound pressure level) with a hearing aid before surgery or with the CI speech processor after surgery. The objective audiological assessment was performed before surgery using auditory brainstem responses (ABR). In case of negative ABR in the affected ear, an in-house subjective hearing nerve test was performed. During that test, an electrode is placed in the external auditory meatus, and electrical stimulation is used to determine the integrity of the auditory nerve. During the stimulation, rectangular alternating current electrical stimulation is applied for 500 ms. The three frequencies used are 62 Hz, 125 Hz, 250 Hz. The stimulation begins at 100 μA, and the current is increased manually up to 1000 μA. In most patients with preserved residual hearing, stimulation with 200–300 μA induces an acoustic sensation of humming. The amperage causing the hearing sensation is designated as the electrical hearing threshold (EHT). After defining the EHT, the current is increased further until an intense, unpleasant sensation of a sound occurs (designated unpleasant hearing level—UHL). The difference between the EHT and the UHL is called dynamic.

Oldenburger Sentence Test (OLSA) was used to assess hearing with background noise after implantation [30]. OLSA consists of a five-word sentence and is defined as a signal-to-noise ratio in dB speech pressure level (SPL). OLSA was performed with CI processor switched on and masking of the other ear. Speech and noise were presented from the front. Additionally, the bilateral hearing was tested with CI and hearing aid (when a patient had a contralateral hearing aid).

Radiological diagnostics included cranial computed tomography (CT) of the temporal bone, and magnetic resonance imaging (cMRI) acquired with a 1-mm slice thickness using intravenous gadolinium contrast. The MRI protocol included T1- and T2-weighted sequences, T1 VIBE (volumetric interpolated breath-hold examination), fat-saturated post gadolinium sequences, and T2w SPACE sequences and/or 3D CISS (constructive interference in steady-state) sequence. The cMRI-imaging identified the VS in all cases. According to the Samii [30] grading system, the tumors were classified as T1 or T2, indicating the extension in the internal auditory meatus (T1) or the extension to the cerebellopontine angle with additional extrameatal growth. 

## 3. Results

Patients # 1–3 were incidentally diagnosed with VS during pre-implantation diagnostics. All patients experienced progressive hearing loss, resulting in profound sensorineural hearing loss or deafness in the VS-affected ear. Patients # 1, 3, and 4 were bilaterally deaf, whereas Patient #2 had asymmetric hearing loss (see Table 1). All patients reported episodes of vertigo; two patients (#1 and #4) had a total loss of vestibular function in the ear affected by VS, and the other patients (#2 and #3) had a compensated partial loss of the vestibular function in the VS ear. Patient #1 had tinnitus causing mild distress with good compensation strategies. Table 1 summarizes the demographic data.

After discussing the clinical findings and treatment options during the interdisciplinary Skull Base Board meeting and with the patients, three patients (# 1, 3, and 4) opted for primary CyberKnife radiosurgery. Their reasons for deciding against the microsurgical removal were advanced age, comorbidities and/or small tumor size. Patient # 2 decided for secondary Cyberknife treatment for residual VS after initial surgical treatment. 

Unilateral implantation was performed in all patients after CyberKnife radiosurgery. The following cochlear implants were used: (CI) model CI 512 (patient #1), MedEl Synchrony Flex 28 (patients # 2 and 4), and Advanced Bionics Hires Ultra 90K (patient # 3). The implantation procedure was successful in all patients; no symptoms of cochlear fibrosis or post-radiation changes in the middle and inner ear after SRS were observed. 

The mean preoperative aided WRS with a hearing aid at 65 dB (FMT) was 5.0 ± 10% (range 0–20%, *n* = 4). Six months postoperatively, the mean aided WRS (65 dB, FMT) with CI was 60.0 ± 22.7% (range 35–90%, *n* = 4). One year postoperatively, the mean aided WRS (65 dB, FMT) with CI was 46.7 ± 12.6% (range, 35–60%, *n* = 3, one-year follow-up for Patient #4 is not yet available). Table 2 summarizes treatment and audiology outcomes.

### 3.1. Case 1

The 63-year-old patient sought consultation primarily for progressive hearing loss on both sides for approximately ten years. PTA showed a bilateral profound hearing loss (see Figure 1), and FMT revealed speech perception (at 65 dB with hearing aids) of 0% on the right side and 10% on the left side. 

Cranial MRI visualized a small 2-mm diameter IC VS on the right side. The following treatment options for VS were recommended: watch-and-scan (every six months) or CyberKnife treatment, cMRI to verify tumor control, and sequential CI surgery. The patient chose the second option and underwent CyberKnife SRS (13 Gy, 85% Isodose; Dmax 15.3 Gy) for the right ear affected by the tumor. Six weeks after SRS, cochlear implantation (CI 512) was performed on the contralateral left ear. Two years later, after cMRI demonstrating the unchanged status of VS, the patient was scheduled for CI surgery (CI 512) on the right side. Hearing rehabilitation was successful, and six months after implantation, the patient achieved an aided WRS of 55% (FMT at 65 dB) and a number recognition score (two-digit numbers, Freiburg Number Test, FNT) of 90% on the right side. One year after the second cochlear implantation, the WRS was 75% (FMT at 65 dB) with binaural CI. The number recognition score (FNT at 65 dB) with binaural CI was 100% (6 months and one year postoperatively).

Hearing in noise was measured with the Oldenburg Sentence test (OLSA) with noise and speech presented from the front. Six months postoperatively, this test revealed scores of −0.3 dB signal-to-noise ratio (SNR) for the right side and −3.7 dB with bilateral CI, indicating an improvement of hearing in noise with CI.

### 3.2. Case 2

A 61-year-old male patient was referred for hearing rehabilitation with CI on the left side because of progressive asymmetric hearing loss (see Figure 2) and limited communication ability, restricting his professional performance as a dentist. WRS on the left side with a hearing aid was 20%, and on the right side, 50% (FMT at 65 dB SPL).

Contrast-enhanced cMRI revealed an intra- and extracanalicular VS (Samii T2) (Figure 3A,B). Possible treatment options before cochlear implantation included tumor removal via a retrosigmoid approach or SRS as first-line therapy. The patient decided on the first option. Intraoperatively, the vestibulocochlear and facial nerves were preserved as the functionality was monitored with electrophysiologic monitoring (neuromonitoring). cMRI demonstrated a small residual IC VS 6 months postoperatively, and the patient decided to undergo SRS before the cochlear implantation. The SRS was performed during a single session with 13 Gy (70% Isodose; Dmax 18.6 Gy). The implantation was performed successfully six weeks later. The CI was placed more posterior than usual to minimize the artifacts in postoperative MRI scans [31]. Six months after CI surgery, the first postoperative MRI was performed using 1.5 Tesla MR with medium bandwidth (see Figure 3C,D for post-CI MRI; the VS is marked with green arrows). Figure 4A,B demonstrates the patient with the Rondo 2 speech processor (images used with the patient’s approval).

One month after implantation, aided WRS with CI on the left side (and masking of the right side) was 45% (FMT at 65 dB SPL) and six months later, 60%. Binaural hearing with CI on the left side and hearing aid on the right side resulted in an aided WRS of 90% (FMT) with CI only after two years.

Hearing in noise was measured with the aided Oldenburg Sentence test (OLSA). One year postoperatively, this test revealed scores of −1.1 dB signal-to-noise ratio (SNR) for the left side with CI (with masking of the right ear) and −3 dB with CI and hearing aid on the right side, indicating an improvement of hearing in noise with CI.

### 3.3. Case 3

A 76-year-old male complained of bilateral progressive hearing loss for approximately 35 years (see Figure 5) and recurrent acute hearing loss on both sides. He reported no tinnitus or vertigo. The patient had a profound hearing loss on the left side with an aided WRS of 20% at 65 dB SPL (FMT) and 0% on the right side with bilateral hearing aids. The hearing nerve integrity was tested with an electrode in the external auditory meatus; the patient could hear humming when the amperage of 531 µA was applied.

Cranial MRI performed during evaluation for implantation revealed a multilocular schwannoma on the right side: small IC VS (T1) and a small intracochlear schwannoma (Figure 6A,B). The case was discussed during the meeting of the Interdisciplinary Skull Base Board. The debated tumor treatment options included resection with a translabyrinthine approach, CyberKnife radiosurgery, or watch-and-scan. All three options were proposed and explained in detail to the patient. In addition, the patient was offered cochlear implantation on the contralateral ear with residual hearing. After presenting possible therapy options for tumor treatment and auditory rehabilitation with CI, the patient decided to treat both tumors using CyberKnife radiosurgery (13 Gy, 70% Isodose; Dmax 18.6 Gy) and opted out from cochlear implantation on the contralateral left ear. One and a half years after the CyberKnife treatment, following two cMRI examinations demonstrating stable tumor (Figure 6D), the patient opted for cochlear implantation on the right side. Two years later, as he was very satisfied with the right ear’s auditory outcome, he opted for CI on the left ear. Twelve months postoperatively, the patient had an aided WRS of 35% (FMT) on the right side. He uses the CI over 10 h daily and has received the second CI two years after the first one. One year postoperatively, OLSA test revealed a 4.6 dB signal-to-noise ratio (SNR) with CI on the right side and hearing aid on the left side and 11.3 dB for the right side with CI (with masking of the left ear).

### 3.4. Case 4

A 57-year-old female patient presented with IC VS on the right side with profound hearing loss after SRS. At the age of 4, she had mumps resulting in a profound sensorineural hearing loss on the left side. An earlier CI evaluation revealed a negative promontory test on the left side. In 2010, she developed hearing loss on the right side, and IC VS was detected using cMRI. SRS was performed in 2019 in a different hospital (3 × 6 Gy) to stop tumor progression and prevent further hearing loss (Figure 7A: pre-therapeutic PTA). Unfortunately, the hearing loss progressed (Figure 7B), and by August 2020, aided WRS with a hearing aid was 0% on the right side. Therefore, after cMRI demonstrated a stable tumor, the patient decided on hearing rehabilitation with CI on the right side. The implantation was performed in our unit in November 2020 without complications. Two months after CI, aided PTA improved remarkably (Figure 7C), and the patient understood 90% of the monosyllables at 65 dB (FMT), remaining on that level six months after implantation.

Hearing in noise was postoperatively measured with the aided Oldenburg Sentence test (OLSA). One year postoperatively, the patient scored 1.5 dB signal-to-noise ratio (SNR) with unilateral CI. 

## 4. Discussion

This study presents four sporadic VS cases and demonstrates that the innovative concept of SRS and CI for patients with sporadic VS and hearing loss is feasible and safe and that the auditory performance of implanted patients improves significantly. This concept cannot only be used for the therapy of small intracanalicular VS (patient #1), but also patients with surgically removed VS (patient #2), with multilocular VS and intracochlear location of VS (patient #3), or progressive hearing loss after SRS (patient #4). 

The use of SRS for VS therapy increases [11,13]. In patients with small- to mid-sized schwannomas and IC location, it has proved to be a safe and minimally invasive method compared to the surgical removal of the tumor. An advantage of SRS is a reasonable and stable local tumor control [14]. To date, only a few reports about patients with neurofibromatosis Type 2 (NF2) and sporadic VS who underwent SRS treatment and hearing rehabilitation with CI have been published so far (see Table 3). Lustig et al. [24] were the first to describe CI in bilaterally deaf NF2 patients with bilateral VS. Two patients underwent Gamma Knife RS for VS treatment and sequential CI with improved post-treatment WRS. Tran et al. [32] reported a patient with NF2 and bilateral VS who underwent Gamma Knife RS for a grade I VS on the right side and surgery for a grade III VS on the left side. Two years later, the patient lost his residual hearing on the right side. He underwent sequential cochlear implantation and hearing rehabilitation with CI, resulting in a WRS of 96% (open-set word score) 1 year postoperatively and 100% four years postoperatively. Trotter and Briggs [33] described three cases with NF2, where the treatment included fractionated radiation therapy and hearing rehabilitation with CI. Two patients had profound sensorineural hearing loss before radiation therapy, and one patient hereafter. Still, the radiation dose in the latter was higher (54 Gy in 30 fractions) than the doses in our study (13 Gy). Hearing results in all patients were good and comparable to our results. Carlson et al. (34) reported four patients with NF2 treated with SRS with doses of 13–20 Gy, who underwent sequential CI 1 to 235 months later. Interestingly, one patient in that study experienced progressive tumor growth 2.5 years after SRS and cochlear implantation. Despite tumor growth, auditory rehabilitation was beneficial. The patient underwent CI explantation and tumor resection; later, he was implanted with a brainstem implant.

Amoodi et al. [26], Costello et al. [32], and Pisa et al. [29] also reported SRS treatment and successful hearing rehabilitation with CI in patients diagnosed with NF 2-associated VS. Deep et al. [36] described NF2 patients either treated with microsurgical resection, SRS, or observational tumor management who underwent cochlear implantation. Patients with SRS or observational management had better audiological performance after cochlear implantation than the CI patients with microsurgical removal of the VS. Urban et al. [28] also outlined the possibility of conservative observational management of the VS and cochlear implantation and reported there better open set perception than in the irradiated ears. Considering these results, the option of observing the VS combined with cochlear implantation needs to be discussed with patients as an alternative to SRS when designing a treatment plan. Patel et al. [27] delineated in a recently published study that patients with NF2-associated VS and sporadic VS benefit from CI after SRS treatment, indicating the possibility of an immediate or delayed CI after SRS. 

In our present study, auditory rehabilitation with CI was successful in all patients, and the mean WRS (at 65 dB SPL, FMT) improved from 5.0 ± 10% before implantation to 60.0 ± 22.7% six months postoperatively. Patient #4 scored 90% in the speech recognition test of the monosyllabic words at 65 dB (FMT). Interestingly, hearing rehabilitation with CI seems very effective after SRS with preserved cochlear nerve, and hearing outcomes are comparable to non-irradiated CI ears. The postoperative aided WRS in the discussed studies ranged from 28% to 96%. Therefore, we assume that SRS might not necessarily damage the cochlear nerve fibers, and the auditory nerve is still functional. The degree of damage to the cochlea and the cochlear nerve resulting from SRS and their long-term consequences are still not fully understood. However, one may assume that essential prognostic factors for good postoperative hearing rehabilitation and WRS include a short duration of deafness or residual hearing and preserved cochlear anatomy without fibrosis. 

### 4.1. Cochlear Changes in Ears with Vestibular Schwannoma

The pathomechanisms of hearing loss in patients with VS are still not fully understood. Still, it is essential to take them into account to understand why cochlear implantation should be considered for patients with VS and NF 2. Various theories explain the pathomechanisms of hearing loss accompanying these diseases, and multifactorial etiology is likely of importance there. One explanation of hearing loss could be tumor-compromised perfusion of the inner ear tissues by a labyrinthine artery. In addition, there is evidence suggesting the occurrence of cochlear changes in patients with VS, such as pathological otoacoustic emissions [33] and histopathological abnormalities. Silverstein [34] found a higher number of proteins in the perilymph of patients with VS than in the perilymph from otosclerosis or Meniere disease patients. However, that study was published in 1971, and it would be of value to repeat it now, using, for instance, mass spectrometry. In their histopathological examination, Roosli et al. [19] detected degeneration of the inner and outer hair cells, stria vascularis, and cochlear neurons in most ears affected by VS compared to unaffected ears. That observation is of great value; however, the temporal bones examined were obtained from persons advanced in age, who lived with the conditions for several years or even decades. Lastly, Mahmud et al. [18] reported eosinophilic precipitates and endolymphatic hydrops in the inner ear affected by VS, adding to the complexity of the disease.

Furthermore, there are theories about cytokines and TNFα causing SHL in the ears with VS [35]. Dilwali et al. [36] found that tumor secretions containing TNFα led to damage to cochlear structures and that there was a correlation between the serum TNFα level and the degree of hearing loss in patients with VS. These theories underline the possibility of hearing rehabilitation with CI in deaf patients with VS despite tumor location on the vestibulocochlear nerve. It appears to be the best option for auditory rehabilitation if a cochlear nerve is preserved based on residual hearing or a positive promontory test in the affected ear.

### 4.2. Hypotheses about Hearing Loss Pathogenesis after SRS

Despite the theories of cochlear changes in ears with VS, it is crucial to remember that SRS may also cause damage to the cochlear structures and the cochlear nerve. In particular, high radiation doses seem to be a risk factor. There are reports of a higher risk of hearing loss after SRS with higher radiation doses and larger irradiated cochlear volumes [37]. The cochlea is radiosensitive and, after radiation doses exceeding 4 Gy, outer hair cell damage and damage to the stria vascularis may occur [38]. In the present study, CyberKnife SRS was performed with 13 Gy (respectively 3 × 6 Gy) as this dose is adequate for local tumor control and allows preservation of the structures mentioned and is in agreement with the findings of Windisch et al. [14]. They treated more than 1000 VS cases with radiation doses of 12.5–13.5 Gy. In this study, we report on a patient (#4) with progressive sensorineural hearing loss before and after SRS. Six months after implantation, he scored 90% of the monosyllables at 65 dB (FMT) with a speech processor. This impressive improvement in speech perception provides evidence for Cyberknife SRS not necessarily damaging the cochlear nerve. It also justifies using cochlear implantation even in cases of sensorineural hearing loss after radiosurgery.

### 4.3. Local Tumor Control after SRS and Alternative Treatment Options

After CyberKnife radiosurgery for VS, local tumor control is estimated to be >95% two years after treatment and >90% after ten years post-treatment [14,39]. In sporadic cases of tumor progression, there is a possibility of re-treatment with SRS [40] or surgical tumor removal. Moreover, radiation therapy is still possible in patients with CI without damaging the implant, as demonstrated by Klenzner et al. [41]. Despite the excellent outcomes of SRS, patients have to be informed about the option of observation of VS and CI. As mentioned above and delineated in earlier studies, patients may be implanted despite the observation management of VS with good audiological outcomes. Another alternative is the microsurgical resection of VS and CI, and—as demonstrated in the present study (patient #2)—a combination of surgery and Cyberknife treatment and sequential CI is feasible and may promote functional auditory outcomes. 

### 4.4. Study Limitations

The major limitations of our study include a small number of patients, the limited follow-up time of the auditory outcome of 6 (*n* = 1), 12 (*n* = 2), and 24 (*n* = 1) months postoperatively, and heterogeneity of cases regarding tumor size and location. The small number of patients is due to the limited time window in which this innovative concept of SRS and cochlear implantation was performed. Moreover, only bilaterally hearing-impaired patients were included. There are few published cases of NF 2, SRS, and CI in the literature and even fewer sporadic VS, SRS, and CI cases. Therefore, this study aims to create awareness for this particular topic and to add to the knowledge of the earlier published cases. Further knowledge should be generated by extending the sample size and designing a multicenter study.

### 4.5. Outlook

There is still a lack of information regarding the long-term outcome of SRS, cochlear implantation, and the degree of neurodegeneration caused by the VS and its treatment. However, especially in patients with bilateral deafness or NF2, the possibility of cochlear implantation paves the path for auditory rehabilitation and improved word recognition compared to the option of an auditory brainstem implant (ABI). Additionally, improved communication is a significant advantage, even though it is unknown how long the hearing nerve will remain functional and if the patient can still hear with a CI after many years or decades. Therefore it would be interesting to perform a long-term follow-up study regarding this topic.

## 5. Conclusions

Cochlear implantation offers an option for successful hearing rehabilitation in selected patients with profound hearing loss associated with vestibular schwannoma or induced by stereotactic radiosurgery of vestibular schwannoma. 

## Figures and Tables

**Figure 1 brainsci-11-01646-f001:**
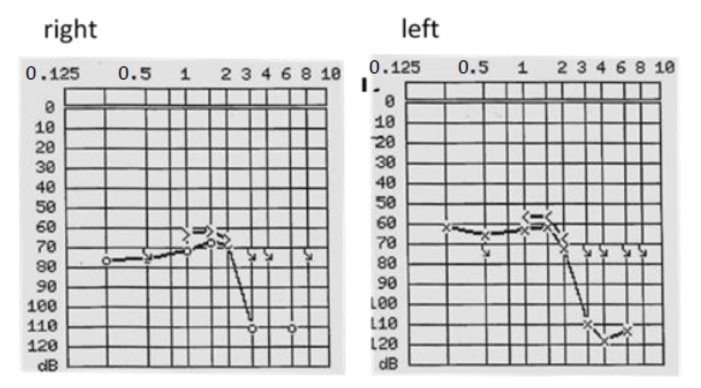
Preoperative PTA of patient #1, which shows a profound sensorineural hearing loss on the right and left side. The frequency on the x-axis is measured in kilohertz (kHZ), the sound pressure level on the y-axis is measured in decibels (dB).

**Figure 2 brainsci-11-01646-f002:**
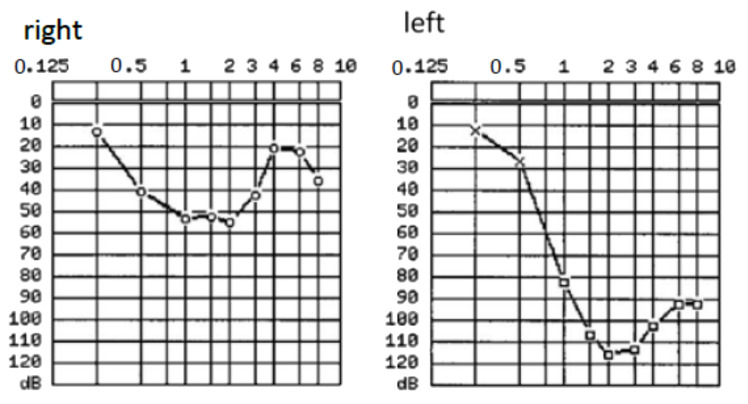
Preoperative PTA of patient #2, which indicates asymmetric hearing loss with mild to moderate sensorineural hearing loss on the right side and mild to profound hearing loss on the left side. The frequency on the x-axis is measured in kilohertz (kHZ), the sound pressure level on the y-axis is measured in decibels (dB).

**Figure 3 brainsci-11-01646-f003:**
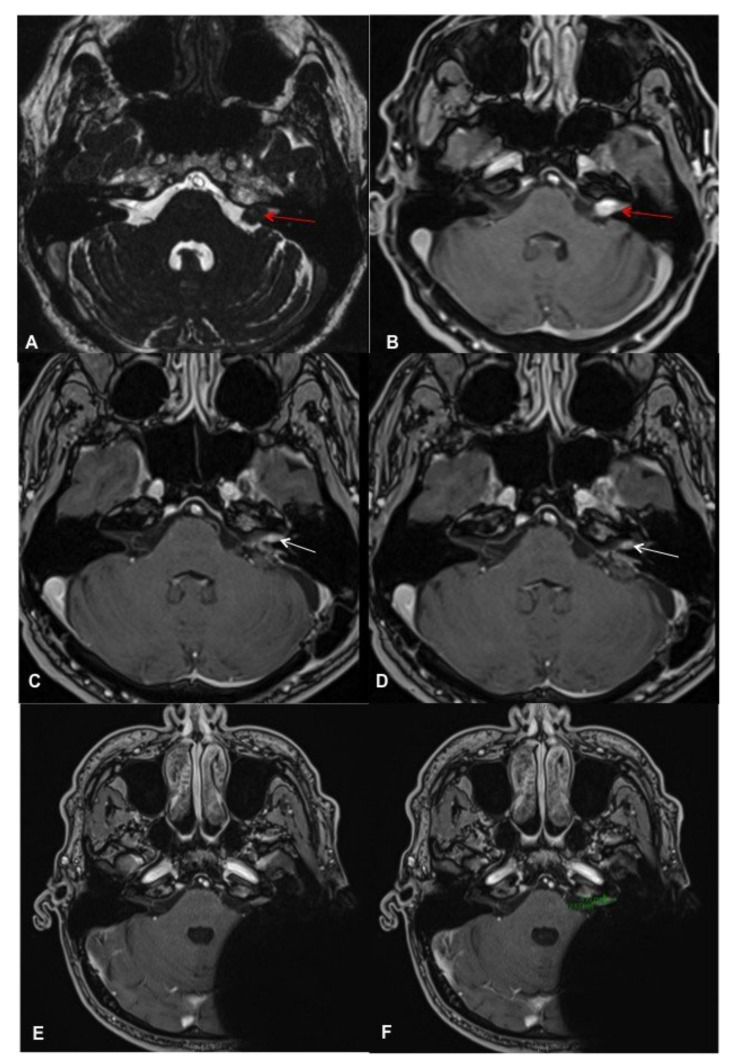
cMRI of patient #2 during treatment. (**A**): T2 CISS (constructive interference in steady-state) sequence; (**B**) T1 VIBE (volumetric interpolated breath-hold examination) sequence. In each panel, the red arrow indicates the intrameatal VS extending to the cerebellopontine angle. (**C**,**D**): T1 VIBE (volumetric interpolated breath-hold examination) sequence. Postoperative cMRI after retrosigmoidal removal of the VS. The white arrow in each panel indicates the residual VS. (**E**,**F**): T1 VIBE after CI. The green line (on panel F) marks the VS diameter after CyberKnife treatment. The filled circle marks the area of extinction by the CI magnet.

**Figure 4 brainsci-11-01646-f004:**
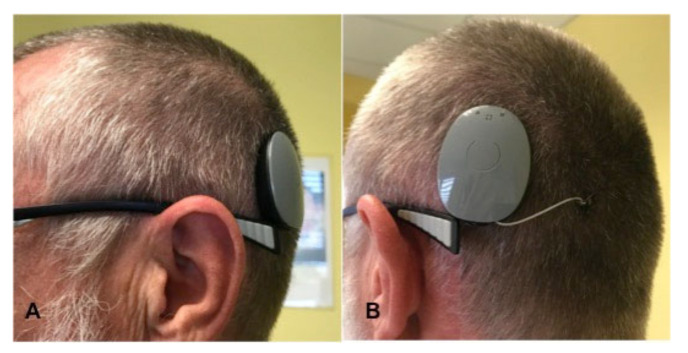
Patient #2 with the Rondo Speech Processor (**A**): profile (**B**): back.

**Figure 5 brainsci-11-01646-f005:**
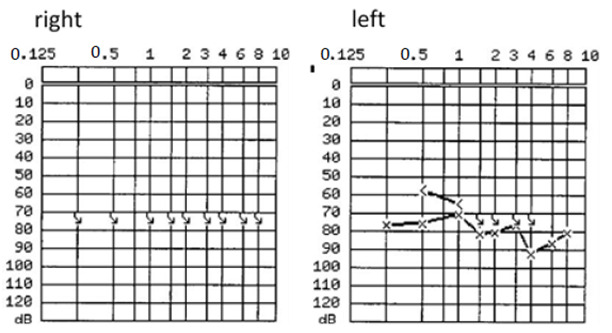
Preoperative PTA of patient #3, which shows deafness on the right side and profound sensorineural hearing loss on the left side. The frequency on the x-axis is measured in kilohertz (kHZ), the sound pressure level on the y-axis is measured in decibels (dB).

**Figure 6 brainsci-11-01646-f006:**
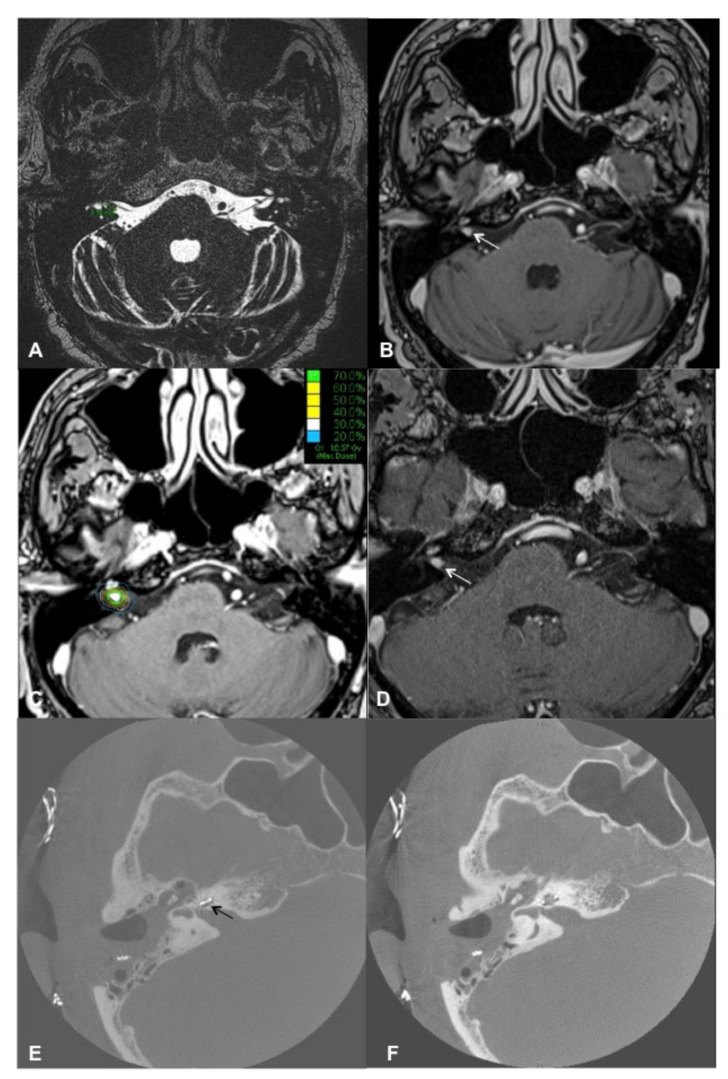
Radiologic images of patient #3. (**A**): T2 CISS (constructive interference in steady-state) sequence: The green lines mark the VS diameter in the preoperative image. (**B**): T1 VIBE (volumetric interpolated breath-hold examination) FS (fat suppression) sequence: Preoperative image, the white arrow indicates the VS. (**C**): Radiation planning images with the lines marking the radiation doses around the VS. (**D**): T1 VIBE FS after CyberKnife treatment: the arrow marks the stable VS. (**E**,**F**): Cone-beam CT after CI: the black arrow in panel E marks the electrode in the basal turn of the cochlea.

**Figure 7 brainsci-11-01646-f007:**
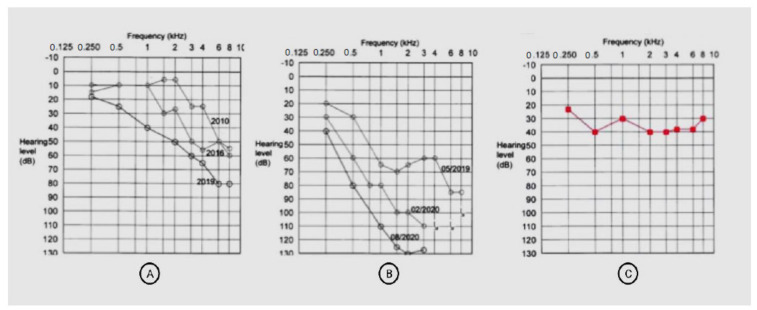
Pure tone audiometry of Patient 4. (**A**): PTA thresholds of the right side in 2010, 2016, and 2019 (before SRS and without hearing aid). (**B**): PTA of the right side after SRS and before CI (without a hearing aid). (**C**): PTA of the right side two months after cochlear implantation with the processor switched on (aided). The frequency on the x-axis is measured in kilohertz (kHZ), the sound pressure level on the y-axis is measured in decibels (dB).

**Table 1 brainsci-11-01646-t001:** Patients’ demographics, tumor and hearing loss characteristics.

Patient Nr.	Gender	Age [Years]	Side of VS (r/l)	VS Localization	VS Size According to Samii	VS Size According to Koos	4-Frequency PTA Threshold	SSD/AHL/DSD
1	m	63	r	IC	T1	1	91.25	DSD
2	m	61	l	IC & CPA	T2	2	87.5	AHL
3	m	79	r	IC	T2	2	no threshold/ deaf	DSD
4	f	58	r	IC	T1	1	103.3	DSD

m = male, f = female, VS = vestibular schwannoma, r = right, l = left, IC = intracanalicular, CPA = cerebellopontine angle, PTA = pure tone audiometry (VS side), SSD = single sided deafness, AHL = asymmetric hearing loss, DSD = double-sided deafness.

**Table 2 brainsci-11-01646-t002:** Vestibular schwannoma location, treatment, cochlear implantation, and outcome.

Patient Nr.	Localization of VS	Therapy of VS	Dose (Gy)	Date of CI	CI Model	WRS Pre CI (%)	WRS 1/2 Months Post CI (%)	WRS Months Post CI (%)	WRS 12 Months Post CI (%)	WRS 24 Months Post CI (%)
1	IC	Cyberknife	13	12/2014	CI 512	0	25	55	45	-
2	intra-& extra-canalicular	retrosigmoidal resection & Cyberknife	13	02/2018	MedEl S. Flex 28	20	45	60	60	55
3	IC	Cyberknife	13	06/2019	AB HiRes Ultra 90 K	0	-	-	35	-
4	IC	Cyberknife	3 × 6	11/2020	MedEl S. Flex 28	0	90	90	-	-

VS = vestibular schwannoma, CI = cochlear implant, IC = intracanalicular, CPA = cerebellopontine angle, WRS = aided word recognition score measured with Freiburg Monosyllabic Tests. The aided WRS was performed with a hearing aid preoperatively and with the CI postoperatively.

**Table 3 brainsci-11-01646-t003:** Literature and case reports covering the topic of radiotherapy or radiosurgery for vestibular schwannoma (VS) and cochlear implantation (CI).

Study	Study Year	Patient Nr	NF 2/Sporadic VS	Radiotherapy/Radiosurgery	Dose in Gray	WRS Pre CI %	WRS Post CI %
Lustig [24]	2006	1 2	NF2 NF 2	RT RT (both GK)	n.k. n.k.	0 0	46(SDS) 46 MTS/0 SDS
Tran [32]	2009	3	NF 2	SRS (GK)	n.k.	deaf	96
Trotter [33]	2010	4 5 6	NF 2 NF 2 NF 2	Frac. RT Frac. RT Frac. RT	54 50.4 12	7 30 n.k.	79 45 n.k.
Carlson [34]	2012	7 8 9 10	NF 2 NF 2 NF 2 NF 2	SRS SRS SRS SRS	13 15 20 16	n.k. n.k. n.k. n.k.	46 86 n.k. n.k.
Amoodi [26]	2013	11	NF 2	SRS	n.k.	0	2
Costello [35]	2015	12	NF 2	SRS (GK)	12.5	0	36 (CUNY)
Pisa [29]	2017	13 14	NF 2 NF 2	SRS SRS (Both GK)	12.5	0 2	52 28
Urban [28]	2020	15–21	NF 2 & sporadic VS	SRS (*n* = 4) Frac. RT (*n* = 1)	n.k.	6	55.7 (mean)
Patel [27]	2021	22–28	NF 2 & sporadic VS	SRS	12–20	0	0–95
Deep [36]	2021	29–34	NF 2	SRS	n.k.	4	30–68
Own study	2021	35 36 37 38	sporadic VS sporadic VS sporadic VS sporadic VS	SRS SRS SRS SRS	13 13 13 3 × 6	0 20 0 0	45 60 35 100

NF2 = neurofibromatosis Type 2, CI = cochlear implant, GK = Gamma Knife, n.k. = not known, WRS = word recognition score, SDS = speech discrimination score, RT = radiotherapy, Frac. = fractionated, SRS = stereotactic radiosurgery, CUNY = City University of New York Sentence Test.

## Data Availability

Data available on request due to privacy and ethical restrictions. The data presented in this study are available on request from the corresponding author. The data are not publicly available due to ethical standards concerning the patients’ privacy.

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
