# Peer review of "Hearing Rehabilitation with Cochlear Implants after CyberKnife Radiosurgery of Vestibular Schwannoma: A Report Based on Four Clinical Cases"

_brainsci, 2021, doi:10.3390/brainsci11121646_

Round 1

Reviewer 1 Report

The article is well constructed and written that provide useful information about CI in VN patients after treatment.

Author Response

Thank you for the feedback!

Reviewer 2 Report

I think the article is of great merit and would be of interest to readers.  But I think there are some additions / qualifications that need to be added. 

  1. It is not clear if the participants are all NF2 - this is implied in the discussion but I dont think this is the case - not sure??  
  2. Specify the aetiology of the hearing loss in the non VS ear.  This has implications for the CI outcome and it is never clear how much of a contribution this has in the VS ear, especially as these examples are co-incidental findings.
  3. Specifiy the degree of hearing loss in the non VS ear - it is generally specified for the VS ear but not the opposite ear.  This is NB as binaural aided WRS is often reported and conclusions based on outcomes of thereof.
  4. Specify the degree of hearing loss in the VS ear.
  5. Distinguish between unaided and aided WRS.  Eg. PTA cannot improve after CI.  
  6.  I suggest that FMT be replaced by WRS as WRS is immediately recognised by the reader, while FMT is not.  Eg. in tables reporting post CI aided WRS scores, replace FMT with WRS rather.  And also in abstract.  Again, distinguishing between aided and unaided WRS is necessary.
  7. Mention is made of a 'hearing nerve test' and at other times promontory stimulation test.  At one point (line 95), hearing nerve test with 'electrode in the external auditory meatus or promontory test' is mentioned.  Clarify please.  Are you talking about two different 'hearing nerve tests' or saying promontory test is performed with the electrode in the external auditory meatus?  I don't think there is a electrically evoked test, eABR, or its subjective counterpart, the promontory test, that is performed with the electrode in the ear canal.  Unless what is being referred to is ABR with response found at maximum output in patient with a severe hearing loss.  When promontory test was specified (eg line 219), was this purely subjective feedback from patient or was there an objective component as measured by the presence of a wave during eABR?
  8. The stimulus level where a 'positive promontory test' / 'positive hearing nerve test' was obtained is relevant.  Elevated mA levels are likely to impact outcomes.  I suggest specifying this.  
  9. There were some strongly worded conclusions at the end, given that this presents evidence from only 4 patients.  I think a more conservative approach would be more appropriate.
  10. What is that in the left ear in the CPA in MRI of patient 3?  It is not in IAM and not VS ... I'm not a surgeon, but thought this would be worth mentioning, irrespective of relevance to CI outcomes?  This links to the clarification of NF2 status of participants....

I have made several more, specific comments in the PDF document attached that require review.

But I enjoyed the article and think with some refinement this will be appropriate for publication, despite the small sample size.

Reviewer 3 Report

The Authors report 4 cases of CI after SRS for small acoustic neuromas confirming the feasibility of CI for an irradiated acoustic nerve.

Actually the manuscript is a case report. I think that material and methods and results chapters are not necessary.

I think that watch and scan should has been the only management option to propose to the patients. Then contralateral CI could be done and when observing no growth sequential CI  performed in the affected side.

Patient 3 "opted" for CI first in irradiated side: why did you give this option to the patient?

Were the audiological tests at diagnosis and at the time of implantation on the affected side unchanged in patients 1, 2, and 3?

I wonder if FMT is sufficient to test CI results

Do you think that SRS could be feasible in a previously implanted ear? In this case you can implant the patient and perform SRS only if the lesion grows, it could be a good option fo NF2 patients.

Reviewer 4 Report

Thank you fro the opportunity to review our article.

The CI implantation after vestibular schwannoma (VS) treatment has been reported before this article. The Cyber Knife or stereotactic removal of VS has also been reported earlier (review Borsetto et al Cochlear Implant int 2020). Although the subject is not entirely novel, the previously published data conserning rehabilitation of hearing with CI in VS induced hearing loss consists small series or case reports, atleast generally. Therefor it is worth to consider this paper for publification, eventhough there is only four cases. 

Couple of minor observations

As this is more of brain sciences and the audiological testing is not probably well known to all of the readers, the used tests could be explained shortly in methods.

There is OLSA results for two of cases postoperatively. Were other two too poor to complete OLSA? Also, OLSA is nt mentioned in the method section, could it be added there? 

Studies with few cases become more beneficial when the results should be presented as versatile as possible. Now there is audiograms for the patient nro 4, could there be audiograms when recordabke for patients 1, 2 and 3 also? When ABR were measured, would it be possible to present the treshold data conserning the ABR? Or was it mostly used to confirm integrity of the cochlear nerve?

Sentence at lines 195-197 refers that the word regocnition was confirmed with ABR. ABR measures the hearing tresholds not the word regocnition.

Round 2

Reviewer 3 Report

When CI is feasible in non-irradiated side it should be implanted first. I think You gave an unethical option to patient 3 and you should say this in the discussion.

Author Response

We appreciate the concern of the Reviewer. However, the way we wrote the paragraph in question left an incorrect impression. We have revised the paragraph and precisely described the decision-making process to address this problem.   Cranial MRI performed during evaluation for implantation revealed a multilocular schwannoma on the right side: small IC VS (T1) and a small intracochlear schwannoma (Figure 5 A&B). The case was discussed during the meeting of the Interdisciplinary Skull Base Board. The debated tumor treatment options included resection with a translabyrinthine approach, CyberKnife radiosurgery, or watch-and-scan. All three options were proposed and explained in detail to the patient. In addition, the patient was offered cochlear implantation on the contralateral ear with residual hearing. After presenting possible therapy options for tumor treatment and auditory rehabilitation with CI, the patient has decided to treat both tumors using CyberKnife radiosurgery (13 Gy, 70% Isodose; Dmax 18.6 Gy) and opted out from cochlear implantation on the contralateral left ear. One and a half years after the CyberKnife treatment, following two cMRI examinations demonstrating stable tumor (Figure 5 D), the patient opted for cochlear implantation on the right side. Two years later, as he was very satisfied with the right ear's auditory outcome, he opted for CI on the left ear.